# Bisphenol BPAF and BPC are agonists for estrogen receptor ERα but antagonists for N-terminal domain-lacking ERα

Xiaohui Liu[1]*, Miki Shimohigashi[2], Yasuyuki Shimohigashi[2,3]*

1 Department of Biotechnology and Life Sciences, Faculty of Biotechnology and Life Sciences, Sojo University, Kumamoto, Japan, 2 Risk Science Research Institute, Fukuoka, Japan, 3 Laboratory of Structure-Function Biochemistry, Department of Chemistry, Faculty of Science, Kyushu University, Fukuoka, Japan

* xiaohui@bio.sojo-u.ac.jp (XL); shimo@kyudai.jp (YS)

## Abstract

In the DNA complex of coactivator-bound estrogen receptor alpha (ERα), the N-terminal domain (NTD) and C-terminal ligand-binding domain (LBD) interact specifically. ERα-NTD and LBD constitute activation function 1 (AF-1) and activation function 2 (AF-2), respectively. We recently revealed that, despite the complete inactivity of ERα-NTD(AF-1), desNTD(AF-1)-ERα exhibited approximately 65% of the activity of the natural estrogen 17β-estradiol (E2) for wild-type full-length ERα. It remains unclear how a deficiency of NTD(AF-1) influences the activity of desNTD(AF-1)-ERα especially with respect to other estrogens and xenoestrogens. The major objective of this study is to evaluate the ligand specificity of desNTD(AF-1)-ERα for a series of xenoestrogens, including bisphenol A (BPA), BPAF, and BPC, together with E2, and the antagonists 4-hydroxytamoxifen (4-OHT) and ICI 182,780 (ICI). The receptors were transiently expressed in HeLa cells, and receptor activation activity was evaluated by luciferase reporter gene assay. Antagonist activity was examined for ERα and desNTD(AF-1)-ERα using E2 as a reference agonist. E2 exhibited full agonist activity for both ERα and desNTD(AF-1)-ERα, whereas 4-OHT and ICI were completely inactive and exhibited antagonist activity for E2 in both ERα and desNTD(AF-1)-ERα. All bisphenols were active for full-length ERα. Surprisingly, however, BPAF and BPC were almost completely inactive for desNTD(AF-1)-ERα, whereas BPA was fully active. BPAF and BPC exhibited distinct antagonist activity for E2 in desNTD(AF-1)-ERα, with $pA_2$ values of 7.62 and 7.86, respectively. The present results revealed that the presence of the N-terminal NTD(AF-1) domain substantiates the agonist activity of halogen-containing BPAF and BPC in wild-type full-length ERα. ERα-NTD(AF-1) plays an essential role in determining the agonist/antagonist activity of BPAF and BPC for estrogen receptor ERα.

**Data availability statement:** All relevant data are within the manuscript and its Supporting Information files.

**Funding:** This work was supported by the Japan Society for the Promotion of Science (JSPS) KAKENHI grant nos. 25K15462, 22K12395, 19K12340, 15K00557, and 25740024 (to X.L.) and 22221005 and 15H01741 (to Y.S.). This work was also supported in part by a Health and Labour Sciences Research Grant for Research on the Risk of Chemical Substances from the Ministry of Health, Labor, and Welfare of Japan, under contract/grant no. H20-Chemistry-General-003 (to Y.S.).

**Competing interests:** The authors declare that no competing interests exist.

**Abbreviations:** AF, activation function; DBD, DNA-binding domain; DMSO, *N,N*-dimethylsulfoxide; DR, dose-ratio; E2, 17β-estradiol; ERα, estrogen receptor alpha; ERE, estrogen response element; 4-OHT, 4-hydroxytamoxifen; GST, glutathione *S*-transferase; H12, α-helix No. 12; ICI, ICI 182,780; LBD, ligand-binding domain; LBP, ligand-binding pocket; NR, nuclear receptor; PDB, protein data bank; SD, standard deviation; SRC-1, steroid receptor coactivator-1; 3D, three-dimensional.

## Introduction

When the quaternary structure of the DNA complex of coactivator-bound estrogen receptor alpha (ERα) was elucidated using cryo-electron microscopy, the adjacency of the N-terminal domain (NTD) and C-terminal ligand-binding domain (LBD) was observed in each ERα protein molecule (Fig 1A) [1]. ERα-NTD and ERα-LBD constitute activation function 1 (AF-1) and activation function 2 (AF-2), respectively, which are essential for transcription activation of wild-type full-length ERα [2,3] (Fig 1A and 1B). This spatial proximity of AF-1 and AF-2 was observed for the first time and judged to be essential for ERα to recruit specific steroid receptor coactivator (SRC) proteins [1,4]. Because NTD and LBD include AF-1 and AF-2, respectively, we refer to them as NTD(AF-1) and LBD(AF-2) henceforth. We term the spatial proximity or interaction between NTD(AF-1) and LBD(AF-2) the N/C-intramolecular interaction.

ERα, a member of the nuclear receptor (NR) superfamily, regulates a number of physiological processes, including reproduction, development, embryogenesis, and metabolism, and is activated by the sex hormone 17β-estradiol (E2) [5–8]. ERα consists of 595 amino acid residues and, like other NRs, has a domain structure comprising an N-terminal NTD(AF-1) domain, a DNA-binding domain (DBD), and a C-terminal LBD(AF-2) domain [9,10]. In wild-type ERα, these domains span residues 1–180 for NTD(AF-1), 181–263 for DBD, and 303–595 for LBD(AF-2), which includes the F-domain (Fig 1B). A flexible hinge region (residues 264–302) connects the DBD and LBD(AF-2). These domains are designated alphabetically as A/B, C, D, E, and F from the N-terminus to the C-terminus (Fig 1B). E2 binds specifically to a site within the ligand-binding pocket (LBP) of the dimeric LBD(AF-2) [11].

desNTD(AF-1)-ERα (residues 181–595) exhibits surprisingly high residual transcriptional activity. Indeed, this NTD(AF-1)-lacking ERα retains approximately 65% of wild-type ERα activity [12], suggesting that wild-type full-length ERα and desNTD(AF-1)-ERα recruit different sets of SRC coactivator proteins. The C-terminal LBD(AF-2) domain appears to exist in different states in wild-type ERα and desNTD(AF-1)-ERα. This difference in the state of LBD(AF-2) likely affects the binding of chemical substances to the LBP of ERα-LBD(AF-2). Therefore, it is important to analyze the activity of a series of estrogens and xenoestrogens [13–15] for both full-length ERα and desNTD(AF-1)-ERα. In this study, the presence or absence of N/C-intramolecular interaction is crucial for evaluating the ligand–receptor interaction.

We previously reported that potent endocrine-disrupting chemicals such as bisphenol AF (BPAF) and bisphenol C (BPC) (Fig 1C), as well as other bisphenols, function as agonists for ERα but as antagonists for ERβ [16–19]. ERβ possesses an NTD(AF-1) domain that is 36 residues shorter than that of ERα-NTD(AF-1). BPAF and BPC were found to bind to ERα and ERβ considerably strongly, and thus they were with a robust concern for their endocrine-disturbing action in the human body [20]. Although the amino acid sequences of the NTD(AF-1) domains are dissimilar, the DBD and LBD(AF-2) domains of ERα and ERβ are highly similar. We hypothesized that the ERα-agonist/ERβ-antagonist activity of BPAF and BPC might be due

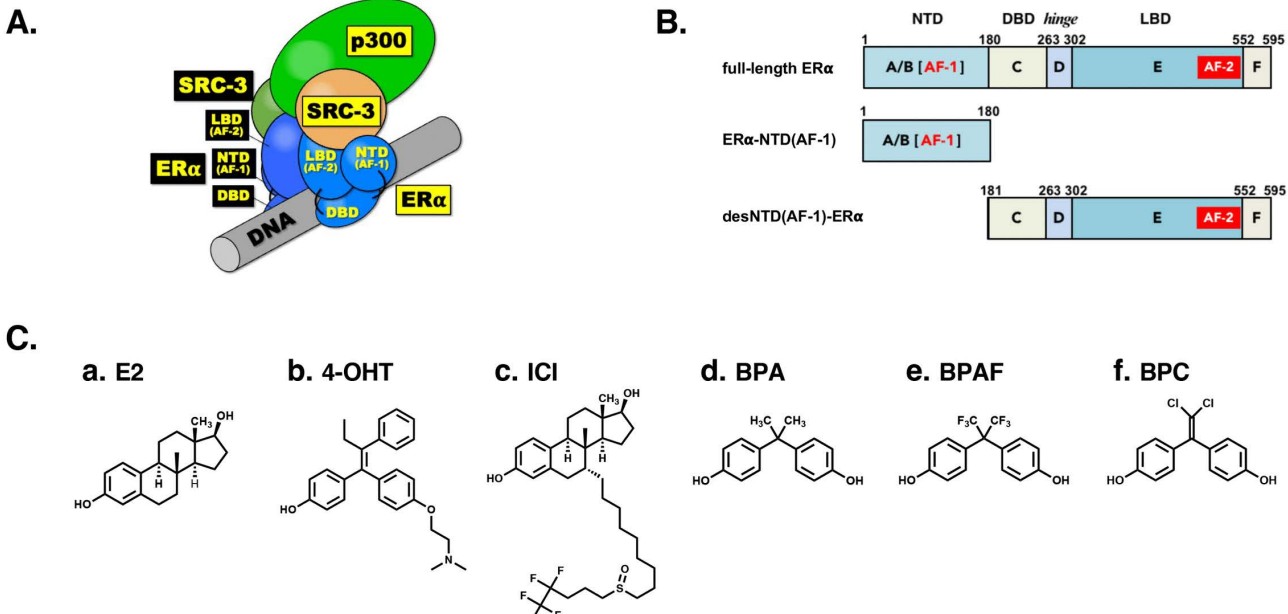

**Fig 1. Structure of estrogen receptor ERα, desNTD(AF-1)-ERα, and their ligands.** (A) Schematic of the structure of the ERα-coactivator (SRC-3, p300)-DNA complex. (B) Schematic bar representation of domain structures of full-length ERα, ERα-NTD(AF-1), and desNTD(AF-1)-ERα. (C) Structures of a series of estrogen receptor ligands: **a.** E2 (17β-estradiol); **b.** 4-OHT (4-hydroxytamoxifen); **c.** ICI (ICI 182,780); **d.** BPA (bisphenol A); **e.** BPAF (bisphenol AF); and **f.** BPC (bisphenol C).

to a difference in the N-terminal NTD(AF-1) domains. This hypothesis prompted us to examine the activity of a series of estrogens and xenoestrogens against desNTD(AF-1)-ERα.

In the present study, we evaluated the intrinsic receptor characteristics of ERα and desNTD(AF-1)-ERα using a traditional agonist and antagonists for wild-type ERα: E2, 4-hydroxytamoxifen (4-OHT), and ICI 182,780 (ICI) (Fig 1C). We also assayed bisphenol A (BPA), a chemical compound primarily used in the manufacture of various plastics. BPA is a xenoestrogen and endocrine disruptor [21] that mimics the effects of estrogen in the body. We found that BPA binds strongly to human estrogen-related receptor gamma (ERRγ), but only weakly to ERα [22,23]. We further demonstrated that ERRγ, on of 48 human nuclear receptors, is most abundantly expressed in the human placenta [24]. In these a few decades, serious concern is the leakage of BPA and its substitutes from chemical products. In particular, a strong apprehension has been raised about crucial endocrine-disrupting effects to human health, especially as the adverse impacts on the development of the reproductive systems and fetal brain. BPAF and BPC bind to ERRγ fairly strongly, and now ERα, ERβ, and ERRγ appear to be the major targets of bisphenols among human endocrine hormone receptors [17,19,25–27]. In the present study, BPA, BPAF and BPC were tested against ERα and desNTD(AF-1)-ERα. We describe the receptor binding affinity, transcriptional activation activity, and antagonist activity of these estrogens and xenoestrogens, and discuss the molecular mechanisms of receptor responses in full-length ERα and desNTD(AF-1)-ERα.

## Materials and methods

### Chemicals

17β-Estradiol (E2; CAS No. 50-28-2; 98.9% purity) was purchased from Research Biochemicals International (Natick, MA, USA). BPA (2,2-bis(4-hydroxyphenyl)propane; CAS No. 80-05-7; >99% purity), BPAF (2,2-bis(4-hydroxyphenyl)hexafluoropropane; CAS No. 1478-61-1; >98% purity), and ICI 182,780 (ICI;

7α,17β-[9-[(4,4,5,5,5-pentafluoropentyl)-sulfinyl]nonyl]estra-1,3,5(10)-triene-3,17-diol; CAS No. 129453-61-8; >98% purity) were purchased from Tokyo Chemical Industry (TCI) Co., Ltd, Tokyo, Japan. BPC (bis(4-hydroxyphenyl)-2,2-dichlorethylene; CAS No. 14868-03-2; 98% purity) and 4-OHT (4-[(Z)-1-{4-[2-(dimethylamino)ethoxy]phenyl}- 2-phenylbut-1-enyl]phenol; CAS No. 68047-06-3; >99.0% purity) were obtained from Sigma-Aldrich (St. Louis, MO, USA).

## Receptor-binding assays for estrogen receptor ERα

*Preparation of GST-fused ERα-LBD protein:* For receptor-binding assays, the glutathione *S*-transferase (GST)-fused ERα-LBD (GST-ERα-LBD) was prepared as previously described [20]. Briefly, a cDNA fragment encoding ERα-LBD (residues 303–595) was amplified by PCR and cloned into the pGEX-6p-1 vector (GE Healthcare BioSciences, Piscataway, NJ, USA) for GST fusion, and the resulting GST-ERα-LBD construct was expressed in *E. coli* BL21α. The receptor protein was purified using an affinity column (10×100 mm) with Glutathione Sepharose 4B (GE Healthcare BioSciences), followed by gel filtration on a Sephadex G-10 column (15×100 mm; GE Healthcare BioSciences). Purity was confirmed by SDS-PAGE using 12.5% polyacrylamide gel and Coomassie brilliant blue staining. Protein concentrations were determined by the Bradford method [28]. The quality of the purified receptor protein was ensured by a saturation binding assay using [³H]E2 (120.5 Ci/mmol; PerkinElmer Life Sciences, Boston, MA, USA) [16,20,29].

*Radioligand binding assay for competitive binding:* The competitive binding assay to measure the binding affinity of a ligand to the ERα-LBD receptor was performed as previously reported [17,19,20]. Briefly, compounds were dissolved in *N,N*-dimethylsulfoxide (DMSO) and serially diluted in binding buffer (50 mM Tris-HCl [pH 7.4], 1 mM EDTA, 1 mM EGTA, 1 mM sodium vanadate(V), 10% glycerol) containing 2 mg/mL γ-globulin using a half-logarithmic 3.16-fold dilution method, maintaining the DMSO concentration below 0.3%. γ-Globulin, but not bovine serum albumin, was used to block nonspecific adsorption to the plasticware. Compounds were evaluated for their ability to inhibit the binding of [³H]E2 (final concentration: 1 nM) to GST-ERα-LBD (60 ng). The assay solutions were incubated at 25°C for 1 h. Bound/free separation was then performed using 100 µl of 0.4% dextran-coated charcoal (Sigma-Aldrich) in PBS (pH 7.4) via direct vacuum filtration using a 96-well filtration plate (MultiScreen^HTS HV, 0.45 µm pore size; Millipore, Billerica, MA, USA) for 2–10 min at 4°C. Radioactivity was determined using a liquid scintillation counter (TopCount NXT; PerkinElmer Life Sciences Japan, Tokyo, Japan).

To estimate binding affinity, $IC_{50}$ (half-maximal inhibitory concentration) values were calculated from dose–response curves generated using GraphPad Prism 10 (GraphPad Software, Inc., Boston, MA, USA). Each assay was performed in duplicate and repeated at least three times.

## *In vitro* biological assays for ERα and desNTD(AF-1)-ERα

*Plasmid construction for ERα and desNTD(AF-1)-ERα:* The expression plasmid of full-length ERα/pcDNA3.1 was generated by PCR amplification of an ERα cDNA clone (OriGene Technologies, Rockville, MD, USA), followed by subcloning into the pcDNA3.1(+) vector (Invitrogen, Carlsbad, CA, USA) as reported previously [16,17]. Using the cDNA clone of full-length ERα as a template, the expression plasmid desNTD(AF-1)-ERα/pcDNA3.1 was similarly constructed by PCR followed by subcloning into pcDNA3.1(+).

*Cell culture and luciferase reporter gene assay:* HeLa cells (RCB0007; RIKEN BRC, Tsukuba, Japan) were seeded at a density of $5 \times 10^5$ cells per well in 6-well microplates and cultured for 24 h at 37°C in a 5% $CO_2$ atmosphere as reported previously [18,19,25–27]. Cells were maintained in modified Eagle's Minimum Essential Medium (Nissui, Tokyo, Japan) supplemented with 10% (v/v) charcoal-treated fetal bovine serum. For transient transfection, three plasmids were co-transfected in a total volume of 2.0 mL medium: (i) the firefly luciferase reporter plasmid pGL3 (Promega, Madison, WI, USA) containing three tandem estrogen response elements (3×ERE/pGL3; ERE=AGGTCAnnnTGACCT) (1500 ng/well), (ii) an expression plasmid full-length ERα/pcDNA3.1 or desNTD(AF-1)-ERα/pcDNA3.1 (500 ng/well), and (iii) the internal control plasmid pGL4.74 (Promega) containing the *Renilla* luciferase gene (100 ng/well) [30]. Transfection was performed

using Lipofectamine™ 3000 (Thermo Fisher Scientific, Waltham, MA, USA) according to the manufacturer's protocol. Following transfection, cells were cultured for an additional 24 h under the same conditions.

For the luciferase reporter gene assay, cells were harvested using 0.05% trypsin/0.02% EDTA solution and plated onto 96-well microplates at a density of $5 \times 10^4$ cells/well. Following compound administration at various concentrations, the cells were incubated for 24 h. Firefly/*Renilla* luciferase activities were measured and normalized using the Dual-Luciferase Reporter Assay System (Promega), and luminescence was measured on a Fluoroskan FL microplate reader (Thermo Fisher Scientific). Each assay was performed in duplicate and repeated independently at least three times. $EC_{50}$ values were estimated from dose–response curves generated using GraphPad Prism 10 (GraphPad Software).

### Luciferase reporter gene assay for transcription inhibitory activity

***Reporter gene assays for measuring antagonist activity:*** For compounds with no transcriptional activity, antagonist activity was initially assessed qualitatively using *in vitro* luciferase reporter gene assay [17,19]. A serial dilution of each compound ($10^{-13}$ to $10^{-5}$ M final concentration) was tested for full-length ERα or desNTD(AF-1)-ERα activity in HeLa cells in the presence of 10 nM E2. This E2 concentration elicits near-maximal activation (approximately 95%) of both full-length ERα and desNTD(AF-1)-ERα. Each assay was performed as described above for transcription activation activity. Apparent $IC_{50}$ values were estimated from dose–response curves generated by GraphPad Prism 10 (GraphPad Software) when activity decreased in a dose–dependent manner.

***Schild plot analysis for antagonist activity:*** For compounds identified as antagonists, inhibitory activity was quantified using Schild plot analysis [31,32]. Seven antagonist concentrations (0.01–10 µM) were prepared via half-logarithmic (3.16-fold) serial dilutions, and a serial dilution of E2 ($10^{-13}$ to $10^{-5}$ M final concentration) was assayed in the presence of each antagonist concentration. The assay was performed twice in duplicate, and data were analyzed from each dose–response curve generated using GraphPad Prism 10 (GraphPad Software). Using the obtained $EC_{50}$ values, the $pA_2$ value, an indicator of antagonist affinity for a receptor, was calculated for each antagonist using Schild plot analysis [32,33].

### Analysis of X-ray crystal structure of ERα-LBD in complex with compounds

Molecular structure information on experimentally determined three-dimensional (3D) structures was downloaded from the RCSB Protein Data Bank (PDB) (https://www.rcsb.org/). All molecular structure studies were carried out using BIOVIA Discovery Studio Visualizer molecular modeling software (Dassault Systèmes, Vélizy-Villacoublay, Paris, France).

### Statistical analysis

All assays were performed at least three times (n ≥ 3) in each independent experiment, with each sample concentration in duplicate. Data analyzed by the software GraphPad Prism 10 are presented as the mean ± standard deviation (SD) for the indicated number of independent experiments. Statistical significance was determined using a two-tailed Student's *t*-test with $p < 0.001$ considered significant.

## Results

### Receptor binding affinity of a series of estrogens and xenoestrogens

Because wild-type full-length ERα and N-terminal NTD(AF-1)-lacking desNTD(AF-1)-ERα share the same ligand-binding domain LBD(AF-2), all ligands would be expected to bind to the ligand-binding pocket (LBP) with the same binding affinity for both full-length ERα and desNTD(AF-1)-ERα. The receptor protein used was GST-ERα-LBD, which possesses a fully open binding pocket. The receptor-binding affinities are shown in Table 1. The natural estrogen agonist E2 exhibited the highest affinity ($IC_{50} = 0.74$ nM) to ERα-LBD, and the synthetic antagonists 4-OHT [34] and ICI [35] showed similar binding affinities (1–3 nM). Among the bisphenols tested, BPC showed the highest affinity (3.02 nM), followed by BPAF and BPA.

**Table 1. Receptor-binding affinity of estrogens and xenoestrogens for ERα.**

| Compounds | Binding affinity[a] | |
| --- | --- | --- |
| | IC$_{50}$ (nM)[b] | Relative affinity |
| E2 (17β-estradiol) | 0.74 ± 0.13 | 100 |
| 4-OHT (4-hydroxytamoxifen) | 1.28 ± 0.22 | 58 |
| ICI (ICI 182,780) | 2.96 ± 0.32 | 25 |
| BPA (bisphenol A) | 1060 ± 78.8 | 0.07 |
| BPAF (bisphenol AF) | 49.6 ± 5.20 | 1.5 |
| BPC (bisphenol C) | 3.02 ± 0.22 | 24 |

[a]The receptor used for the competitive binding assay was GST-ERα-LBD.

[b]The data of half-maximal inhibitory concentration (IC$_{50}$) are presented as the mean ± standard deviation (SD) assessed from at least three independent experiments (n ≥ 3).

### *In vitro* biological activity for ERα and desNTD(AF-1)-ERα

A series of compounds, including E2, 4-OHT, and ICI, were evaluated for their activity on wild-type full-length ERα and N-terminal NTD(AF-1)-lacking desNTD(AF-1)-ERα using a luciferase reporter gene assay. As expected, E2 exhibited strong full transcriptional activation activity (EC$_{50}$ = 0.63 nM) for full-length ERα (Fig 2A) (Table 2). All of the tested bisphenols exhibited full activation activity, as shown in Fig 2A. The order of activity was BPA (EC$_{50}$ = 1,140 nM) <BPAF (67.9 nM) <BPC (4.55 nM) (Table 2). 4-OHT and ICI exhibited no activation activity (Fig 2A).

For desNTD(AF-1)-ERα, E2 exhibited full transcriptional activation activity (EC$_{50}$ = 0.66 nM) (Table 2), but the activity level decreased by approximately 35% (Fig 2B). BPA also exhibited full transcriptional activation activity (1,040 nM) (Table 2), although its activity level decreased by 30–40% (Fig 2Bb). In contrast to BPA, BPAF and BPC were almost completely inactive, showing no transcriptional activation activity (Fig 2Bb). 4-OHT and ICI were also inactive for desNTD(AF-1)-ERα (Fig 2Ba). It is important to note that BPAF and BPC showed inertness or inactivity for desNTD(AF-1)-ERα, which was completely opposite to their full activity for ERα.

### Antagonist activity of compounds inactive for full-length ERα and desNTD(AF-1)-ERα

Compounds inactive for the estrogen receptors may act as antagonists. Therefore, 4-OHT, ICI, BPAF, and BPC were examined for their possible inhibitory activity against the natural estrogen E2. In this study, we tested their effect on the transcription activation activity of E2 at a concentration of 10 nM. This concentration elicits almost full activation (ca. 95%) of ERα or desNTD(AF-1)-ERα. In the presence of the putative antagonists 4-OHT, ICI, BPAF, and BPC, a dose–response curve of 10 nM E2 was generated to assess its residual activity in HeLa cells.

In the cellular experimental condition using full-length ERα, 4-OHT exhibited a distinct antagonist activity for 10 nM E2, reducing the residual activity of 10 nM E2 to baseline (fold-activation = 1; Fig 3Aa). The activity of 10 nM E2 decreased drastically when 4-OHT was administered in the concentration range from $1.0 \times 10^{-13}$ to $1.0 \times 10^{-5}$ M, showing a sharp dose–response curve (Fig 3Aa) and an apparent IC$_{50}$ value of 9.70 nM (Table 3). 4-OHT is acknowledged as a selective estrogen receptor modulator (SERM) that exhibits tissue-specific ERα partial agonist–antagonist activity [36,37]. However, in our luciferase reporter gene assay system using HeLa cells with transient transfections, 4-OHT typically functioned as a pure antagonist through ERα as shown in Fig 3Aa.

ICI was also found to be a pure antagonist for full-length ERα (Fig 3Ba), with an IC$_{50}$ value of 9.92 nM (Table 3). Thus, ICI and 4-OHT were almost equally potent in this assay. ICI itself is reportedly an intrinsically potent pure antagonist with no partial agonist activity [38].

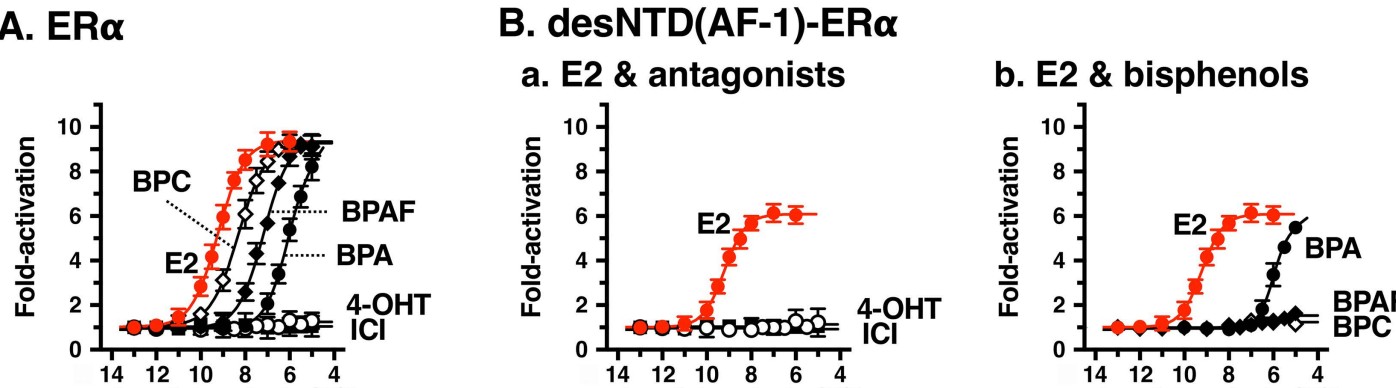

**Fig 2. Transcriptional activity of estrogen and bisphenols for full-length ERα and desNTD(AF-1)-ERα in a luciferase reporter gene assay.**
(**A**) Dose–response curves for full-length ERα: E2 (●), 4-OHT (○), ICI (■), BPA (●), BPAF (♦), and BPC(◇). (**B**) Dose–response curves for desNTD(AF-1)-ERα: **a.** E2 (●), 4-OHT (○), and ICI (■); **b.** E2 (●), BPA (●), BPAF (♦), and BPC (◇). EC$_{50}$ data were estimated as the mean ± SD from at least three independent experiments (n ≥ 3).

**Table 2. Agonistic transcription activation activity for full-length ERα and desNTD(AF-1)-ERα in the luciferase reporter gene assay.**

| Chemicals | Transcriptional activation activity | | | |
|---|---|---|---|---|
| | Full-length ERα | | desNTD(AF-1)-ERα | |
| | EC$_{50}$ (nM) | Relative potency | EC$_{50}$ (nM) | Relative potency |
| E2 | 0.63 ± 0.02[a] | 181,000 | 0.66± 0.07[a] | 173,000 |
| 4-OHT | Inactive[b] | 0 | Inactive[b] | 0 |
| ICI | Iinactive[b] | 0 | Inactive[b] | 0 |
| BPA | 1140 ± 98.7[a] | 100 | 1040 ± 60.9[a] | 110 |
| BPAF | 67.9 ± 7.48[a] | 1,680 | Inactive[b] | 0 |
| BPC | 4.55 ± 0.74[a] | 25,100 | Inactive[b] | 0 |

[a]The values of effective concentration (EC$_{50}$) were calculated from the dose–response curves obtained for transcriptional activation activity. EC$_{50}$ data are presented as the mean ± SD assessed from at least three independent experiments (n ≥ 3).
[b]Almost completely inert, with no measurable transcriptional activation activity.

4-OHT was also a strong antagonist for desNTD(AF-1)-ERα. The dose–response curve depicting the activity of 10 nM E2 showed a clear decrease depending on the doses of 4-OHT, with transcriptional activity being inhibited entirely (the curve reached the basal level, namely, fold-activation = 1) with a 4-OHT concentration of $1.0 \times 10^{-5}$ M (Fig 3Ab). 4-OHT was a fully pure antagonist, with an IC$_{50}$ value of 9.95 nM (Table 3). Similarly, ICI was a fully pure antagonist for desNTD(AF-1)-ERα. The dose–response curve of 10 nM E2 vs. ICI reached baseline (fold-activation = 1; Fig 3Bb), with an IC$_{50}$ value of 10.3 nM (Table 3). Again, ICI and 4-OHT are almost equally potent as antagonists for desNTD(AF-1)-ERα.

### Agonist-to-antagonist functional conversion of BPAF and BPC

For wild-type full-length ERα, BPAF was a fully active agonist. However, in the assay for desNTD(AF-1)-ERα, BPAF became inactive (Fig 2Bb). In the receptor-binding assay to replace [³H]E2 for ERα-LBD, BPAF showed fairly strong binding affinity (IC$_{50}$ = 49.6 nM; Table 1). Despite this, BPAF is inactive for desNTD(AF-1)-ERα, which has the same ligand-binding domain. This suggested that BPAF is a strong antagonist for the NTD-lacking desNTD(AF-1)-ERα,

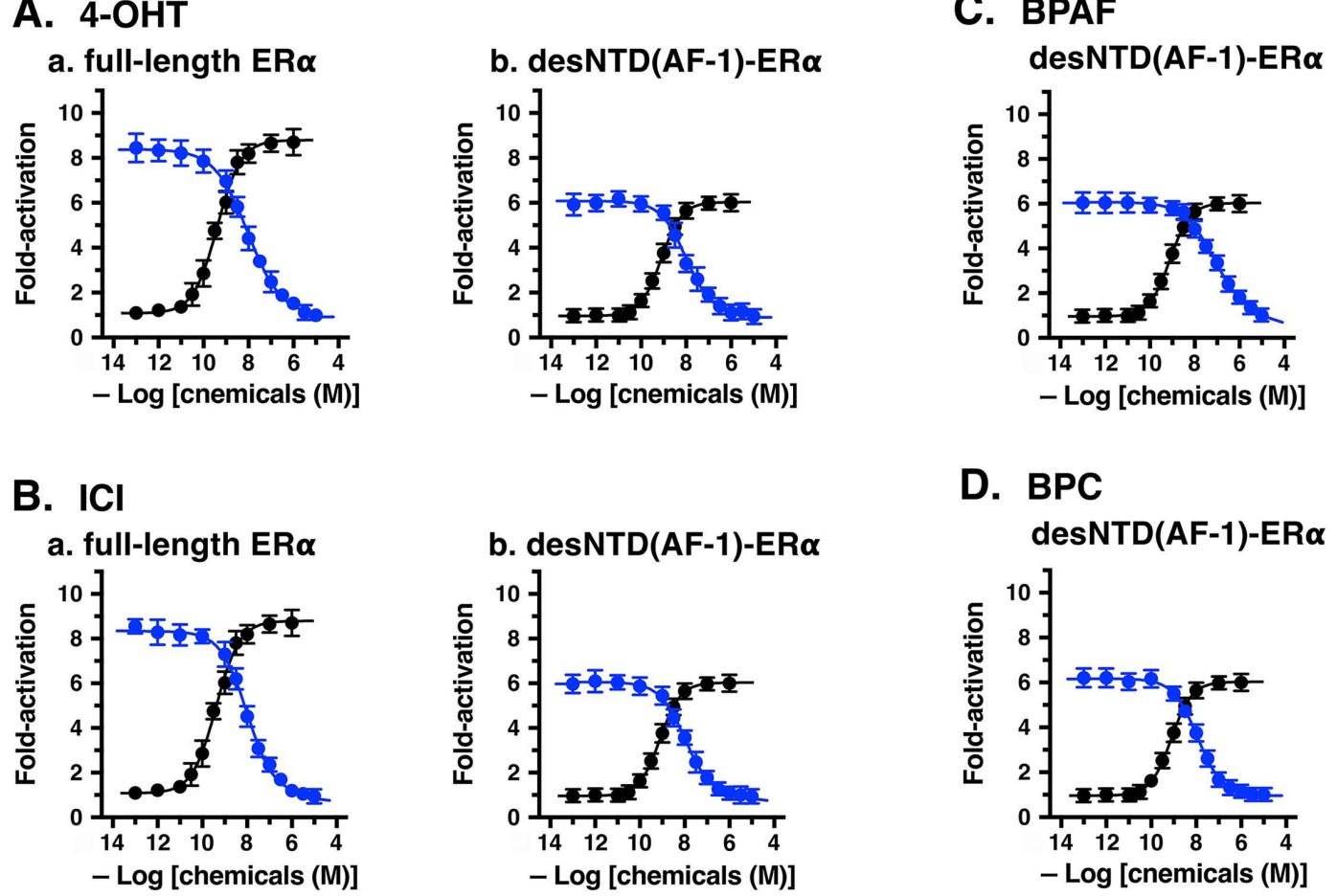

**Fig 3. Transcriptional antagonist activity for full-length ERα and desNTD(AF-1)-ERα in a luciferase reporter gene assay.** Dose–response curves (●) show the transcriptional activation activity of E2 alone for either full-length ERα or desNTD(AF-1)-ERα, while blue curves (●) show the inhibitory activity of each antagonist on 10 nM E2 for the indicated ERα receptor. The antagonistic effects of 4-OHT (**A**) and ICI (**B**) on the agonist activity of 10 nM E2 are shown for full-length ERα (**a**) and desNTD(AF-1)-ERα (**b**), while those of BPAF and BPC are shown only for desNTD(AF-1)-ERα (**C** and **D**, respectively). $IC_{50}$ data were estimated as the mean±SD from at least three independent experiments (n≥3).

competing with E2 for the same ligand-binding site. Indeed, BPAF exhibited a definite inhibitory activity for 10 nM E2 in desNTD(AF-1)-ERα, reducing activity to baseline (fold-activation = 1; Fig 3C). The $IC_{50}$ value of this pure antagonist was 74.8 nM (Table 3).

BPC, like BPAF, was a full agonist for ERα but was inactive for desNTD(AF-1)-ERα. (Fig 2Bb). BPC exhibited a strong binding affinity for ERα-LBD, with an $IC_{50}$ value of 3.02 nM (Table 1), and functioned as an antagonist, as expected (Fig 3D). For desNTD(AF-1)-ERα, BPC potently inhibited 10 nM E2, with its dose–response curve reaching the activity base-line of fold-activation = 1. The $IC_{50}$ value of this pure antagonist was 11.3 nM (Table 2), indicating that BPC is approximately seven times stronger than BPAF as an antagonist.

## Assessment of antagonist activity by Schild plot analysis

4-OHT and ICI were antagonists for both full-length ERα and N-terminal domain-lacking desNTD(AF-1)-ERα (Fig 3A and 3B). Surprisingly, BPAF and BPC were clear antagonists for desNTD(AF-1)-ERα (Fig 3C and 3D), despite their evident

**Table 3. Antagonistic transcriptional inhibitory activity of full-length ERα and desNTD(AF-1)-ERα in a luciferase reporter gene assay.**

| Chemicals | Transcriptional inhibitory activity | | | |
|---|---|---|---|---|
| | Full-length ERα | | desNTD(AF-1)-ERα | |
| | $IC_{50}$ (nM)[a] | $pA_2$[b] | $IC_{50}$ (nM)[a] | $pA_2$[b] |
| E2 | agonist[c] | ———— | agonist[d] | ———— |
| 4-OHT | 9.70 ± 0.95 | 8.02 ± 0.08 | 9.95 ± 0.59 | 8.06 ± 0.14 |
| ICI | 9.92 ± 1.48 | 7.96 ± 0.18 | 10.3 ± 0.82 | 8.01 ± 0.06 |
| BPA | agonist[c] | ———— | agonist[d] | ———— |
| BPAF | agonist[c] | ———— | 74.8 ± 4.71 | 7.62 ± 0.36 |
| BPC | agonist[c] | ———— | 11.3 ± 1.39 | 7.86 ± 0.15 |

[a]The $IC_{50}$ values were estimated from the dose–response curves obtained for transcriptional activation activity in the presence of 10 nM E2. Data are presented as the mean ± SD assessed from at least three independent experiments (n ≥ 3).

[b]The $pA_2$ values were estimated from the Schild plot analysis, and data are as the mean ± SD from at least three independent experiments (n ≥ 3).

[c]Fully potent compound with agonistic transcriptional activation activity.

[d]Fully potent compound, but exhibiting approximately 65% agonistic transcriptional activation activity in full-length ERα.

agonist activity for full-length ERα. To determine their potency as competitive reversible antagonists, Schild plot analysis [31,32] was performed as reported previously [17–19]. This analysis involves measuring the receptor response to an agonist in the presence of varying antagonist concentrations. The $pA_2$ value, determined from a secondary analysis, is then used to calculate the dissociation constant ($K_d$) of a competitive antagonist. The $pA_2$ value measures the affinity of a competitive reversible antagonist for its receptor.

For the Schild plot analysis of desNTD(AF-1)-ERα, BPAF and BPC were tested against the natural estrogen agonist E2. A serial concentration of E2 ($1.0 \times 10^{-13}$ to $1.0 \times 10^{-5}$ M) produced a sigmoidal dose–response curve for its transcriptional activation activity. This curve exhibited successive parallel rightward shifts in the presence of seven different concentrations of BPAF ($1.0 \times 10^{-8}$, $10^{-7.5}$, $10^{-7}$, $10^{-6.5}$, $10^{-6}$, $10^{-5.5}$, and $10^{-5}$ M) (Fig 4Aa) and BPC (Fig 4Ba). The $EC_{50}$ values were estimated from all curves (S1 Table) and used to calculate the dose ratio (DR) for the Schild plot. The $pA_2$ values of BPAF and BPC were assessed from the analyses shown in Fig 4Ab and 4Bb, respectively. The $pA_2$ value of BPAF was 7.62, while that of BPC was 7.86 (Table 3). The $pA_2$ of BPC was slightly higher than that of BPAF, indicating that BPC is a stronger competitive reversible antagonist.

To evaluate antagonist activity, 4-OHT was also examined using full-length ERα and desNTD(AF-1)-ERα (Fig 5). As shown in Fig 5Aa, the sigmoidal dose–response curve of E2 ($1.0 \times 10^{-13}$ to $1.0 \times 10^{-5}$ M) for full-length ERα shifted rightward successively in the presence of increasing concentrations of 4-OHT ($1.0 \times 10^{-8}$, $10^{-7.5}$, $10^{-7}$, $10^{-6.5}$, $10^{-6}$, $10^{-5.5}$, and $10^{-5}$ M). All estimated $EC_{50}$ values (S2 Table) were used to generate a Schild plot, yielding a $pA_2$ value of 8.02 (Fig 5Ab) (Table 3). For desNTD(AF-1)-ERα, successive parallel rightward shifts of E2's dose–response curves were observed with increasing 4-OHT concentrations (Fig 5Ba and S2 Table). Schild plot analysis estimated the $pA_2$ value of 4-OHT to be 8.06 (Fig 5Bb and Table 3). These results indicate that 4-OHT functions as a competitive reversible antagonist for E2 with almost equal potency for both full-length ERα and desNTD(AF-1)-ERα.

ICI was also evaluated for its $pA_2$ value using Schild plot analysis. Fig 6 shows the E2 dose–response curves with successive parallel rightward shifts, and S3 Table shows the estimated $EC_{50}$ values and Log (DR − 1) for both full-length ERα and desNTD(AF-1)-ERα. The $pA_2$ values were 7.96 for full-length ERα and 8.01 for desNTD(AF-1)-ERα (Table 3). These results demonstrate that ICI functions as a competitive reversible antagonist for E2 with almost equal potency for both full-length ERα and desNTD(AF-1)-ERα. Notably, ICI and 4-OHT are almost equipotent antagonists for E2 for both full-length ERα and desNTD(AF-1)-ERα.

## A. BPAF for desNTD(AF-1)-ERα

### a. Dose–response curves

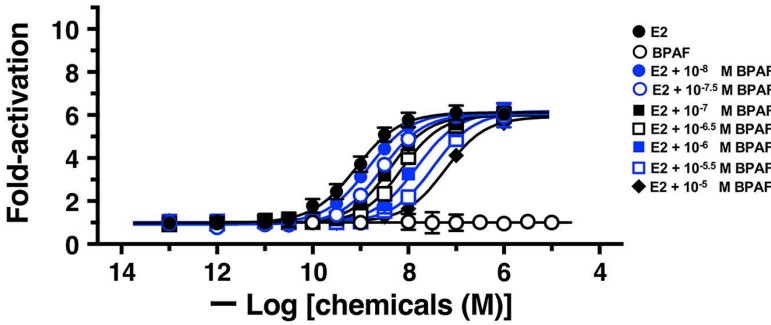

### b. Schild plot analysis

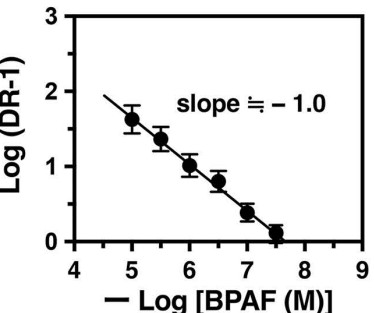

## B. BPC for desNTD(AF-1)-ERα

### a. Dose–response curves

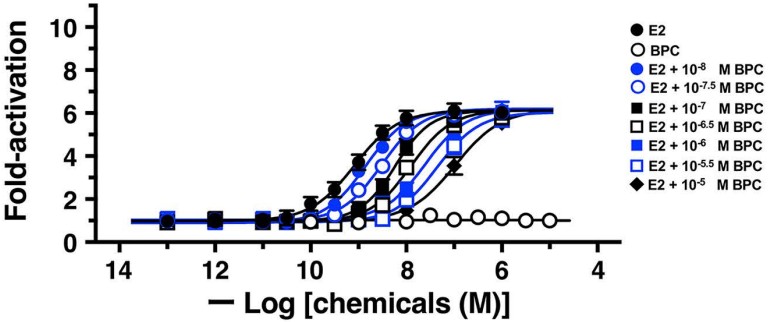

### b. Schild plot analysis

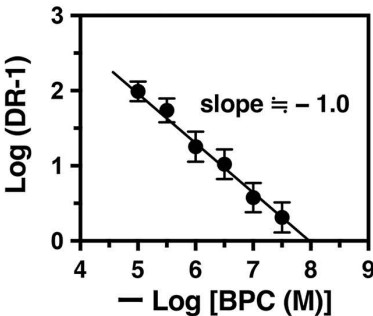

**Fig 4. Detailed analyses of the antagonist activity of bisphenols BPAF and BPC for desNTD(AF-1)-ERα.** (**a**) Dose–response curves of E2 in the presence of seven concentrations ($1.0 \times 10^{-8}$, $10^{-7.5}$, $10^{-7}$, $10^{-6.5}$, $10^{-6}$, $10^{-5.5}$, and $10^{-5}$ M) of BPAF (**A**) and BPC (**B**) for desNTD(AF-1)-ERα, showing successive rightward shifts. (**b**) Schild plot analyses to estimate the $pA_2$ values ($P < 0.0001$) of BPAF (**A**) and BPC (**B**) for desNTD(AF-1)-ERα. $EC_{50}$ data were estimated as the mean ± SD from at least three independent experiments ($n \geq 3$).

## Discussion

### Effect of NTD(AF-1) domain elimination on transcription activity

We confirmed that BPAF and BPC display diametrically opposite effects on transcription activation activity for wild-type full-length ERα and desNTD(AF-1)-ERα. These halogen-containing bisphenols were fully active for full-length ERα but almost completely inactive for desNTD(AF-1)-ERα. Moreover, for N-terminal NTD(AF-1)-lacking desNTD(AF-1)-ERα, BPAF and BPC functioned as competitive reversible antagonists, inhibiting the activity of the natural estrogen E2 in a dose–dependent manner. The functional conversion from agonist to antagonist evidently resulted from the elimination of the N-terminal NTD(AF-1) domain from full-length ERα. This indicates that NTD(AF-1) protects ERα from inactivation by BPAF and BPC.

Both BPAF and BPC bind to the same ligand-binding pocket (LBP) of ERα-LBD(AF-2). These bisphenols were agonists for full-length ERα, but antagonists for desNTD(AF-1)-ERα. Here is a clear functional conversion, 'agonist-to-antagonist' upon the removal of the N-terminal 1–180 peptide fragment containing the activation function motif AF-1. However, the elimination of AF-1 does not appear conclusively important, because the resulting desNTD(AF-1)-ERα remained considerably responsive to natural estrogen E2 (approximately 65% activity remained). LBD(AF-2)-mediated dimerization

## A. 4-OHT for full-length ERα

### a. Dose–response curves

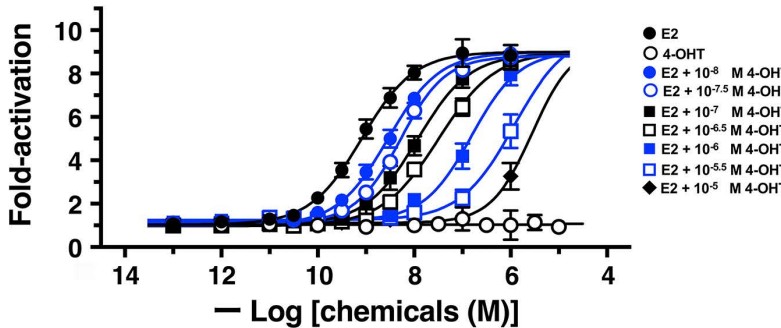

### b. Schild plot analysis

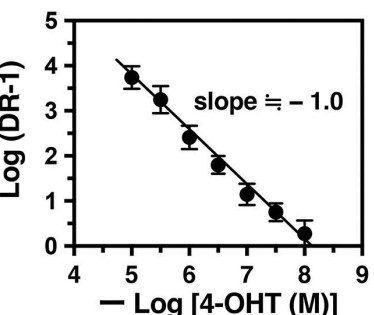

## B. 4-OHT for desNTD(AF-1)-ERα

### a. Dose–response curves

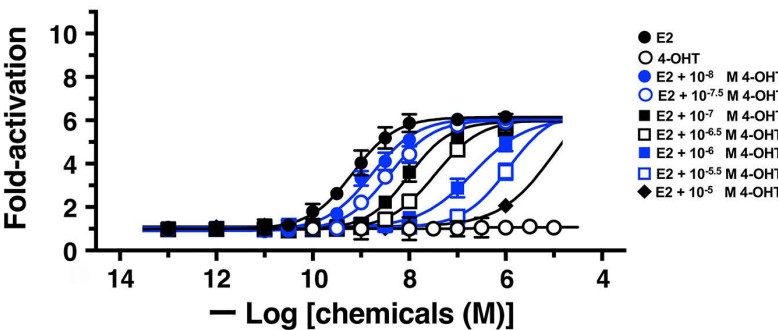

### b. Schild plot analysis

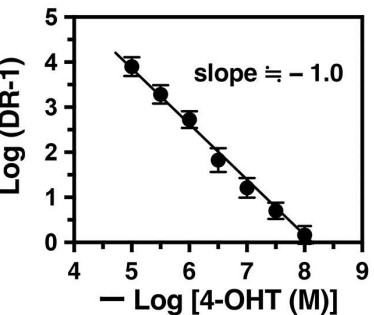

**Fig 5. Schild plot analyses of the antagonist 4-hydroxytamoxifen (4-OHT) for full-length ERα and desNTD(AF-1)-ERα.** Antagonistic activity of 4-OHT for full-length ERα (**A**) and desNTD(AF-1)-ERα (**B**). (**a**) Dose–response curves of E2 in the presence of seven concentrations ($1.0 \times 10^{-8}$, $10^{-7.5}$, $10^{-7}$, $10^{-6.5}$, $10^{-6}$, $10^{-5.5}$, and $10^{-5}$ M) of 4-OHT, showing successive rightward shifts. (**b**) Schild plot analyses to estimate the $pA_2$ values ($P < 0.0001$) of 4-OHT. $EC_{50}$ data were estimated as the mean ± SD from at least three independent experiments ($n \geq 3$).

of full-length ERα recruits a set of coactivators, for instance, two molecules of the p160 family transcriptional coactivator protein SRC-3 and a secondary coactivator p300 [1,4] (Fig 1A). The co-presence of AF-1 and AF-2 is required for this coactivator recruitment; however, even in the absence of the AF-1 motif, desNTD(AF-1)-ERα appears to recruit a distinct set of coactivators to mediate transcription activation.

Regarding the spatial proximity or interaction between NTD(AF-1) and LBD(AF-2), we recently verified such an N/C–intramolecular interaction by 3D structural analysis of full-length ERα on a deep-learning artificial intelligence (AI) system named AlphaFold [12]. From 429 structures in the RCSB Protein Data Bank (PDB), the analysis displayed one particular putative 3D structure (P03372) of wild-type full-length ERα [39]. We found that, in this AlphaFold-predicted ERα structure, the α-helix spanning residues 11–18 in the N-terminal NTD(AF-1) interacts directly with helix 3 (H3) and H12 of the C-terminal LBD(AF-2) domain [12] (Fig 7A). H12 in the LBD(AF-2) domain is itself an AF-2 motif. This N/C-intramolecular interaction was judged essential for wild-type full-length ERα to directly recruit the SRC coactivator proteins [1,12], while desNTD(AF-1)-ERα might fail to recruit the same SRC coactivator protein(s) owing to the lack of the N-terminal NTD(AF-1) domain (Fig 1A).

## A. ICI for full-length ERα
### a. Dose–response curves

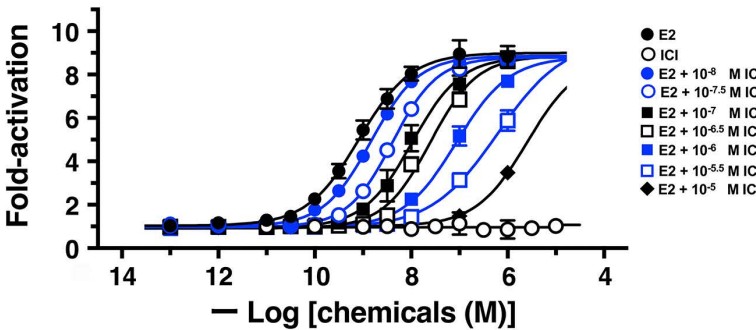

### b. Schild plot analysis

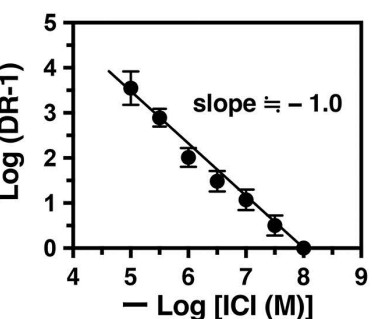

## B. ICI for desNTD(AF-1)-ERα
### a. Dose–response curves

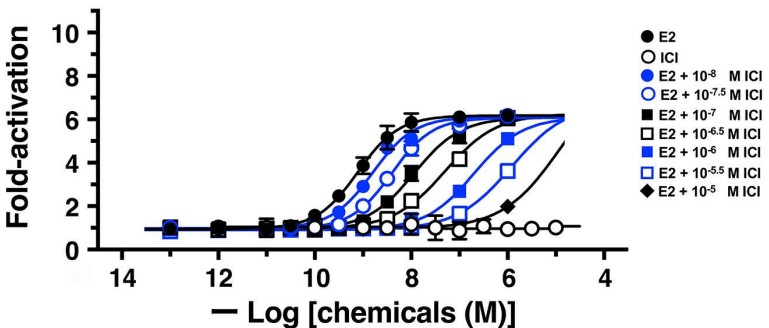

### b. Schild plot analysis

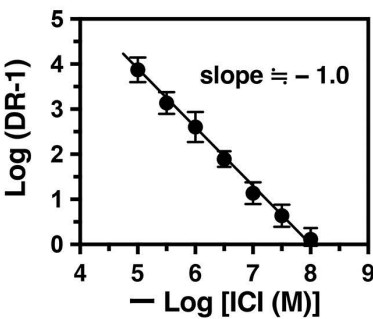

**Fig 6. Schild plot analyses of the antagonist ICI 182,780 (ICI) for full-length ERα and desNTD(AF-1)-ERα.** Antagonistic activity of ICI for full-length ERα (**A**) and desNTD(AF-1)-ERα (**B**). (**a**) Dose–response curves of E2 in the presence of seven concentrations ($1.0 \times 10^{-8}$, $10^{-7.5}$, $10^{-7}$, $10^{-6.5}$, $10^{-6}$, $10^{-5.5}$, and $10^{-5}$ M) of ICI, showing successive rightward shifts. (**b**) Schild plot analyses to estimate the $pA_2$ values (P < 0.0001) of ICI. $EC_{50}$ data were estimated as the mean ± SD from at least three independent experiments (n ≥ 3).

### NTD(AF-1) to LBD(AF-2) interaction holds BPAF and BPC as agonists

To demonstrate the interaction between NTD(AF-1) and LBD(AF-2) (Fig 7A), we examined whether or not untethered free ERα-NTD(AF-1) [residues 1–180 peptide fragment] inhibits the wild-type full-length ERα [12]. If such a synthesized free ERα-NTD(AF-1) and an intact NTD(AF-1) domain in ERα compete for the same ERα-LBD(AF-2) domain in the full-length ERα, severe steric hindrance would have prevented the recruitment of the SRC coactivator proteins. Indeed, synthesized ERα-NTD(AF-1) sharply inhibited the transcription activity of full-length ERα [12].

Additional evidence for the interaction between NTD(AF-1) and LBD(AF-2) was provided by the nuclear localization of NTD(AF-1). When ERα-NTD(AF-1) was expressed solely in HeLa cells, it was found only in the cytoplasm immunocytochemically [12]. However, when expressed together with full-length ERα, ERα-NTD(AF-1) was completely transported from the cytoplasm to the nucleus. This nuclear translocation or transportation was attributable to a direct interaction of ERα-NTD(AF-1) with full-length ERα. ERα, but not ERα-NTD(AF-1), contains the nuclear localization signal [40]. These results were obtained based on the discovery by Yi et al. [1] that ERα-NTD(AF-1) is adjacent to LBD(AF-2) in the ERα•DNA complex (Fig 1A) and on our subsequent speculation that NTD(AF-1) and LBD(AF-2) must interact directly with each other.

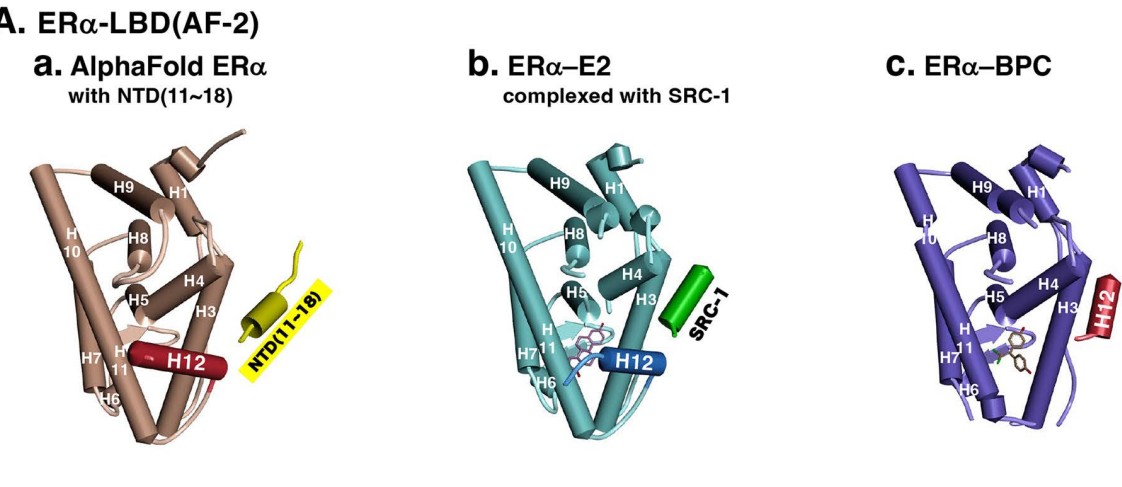

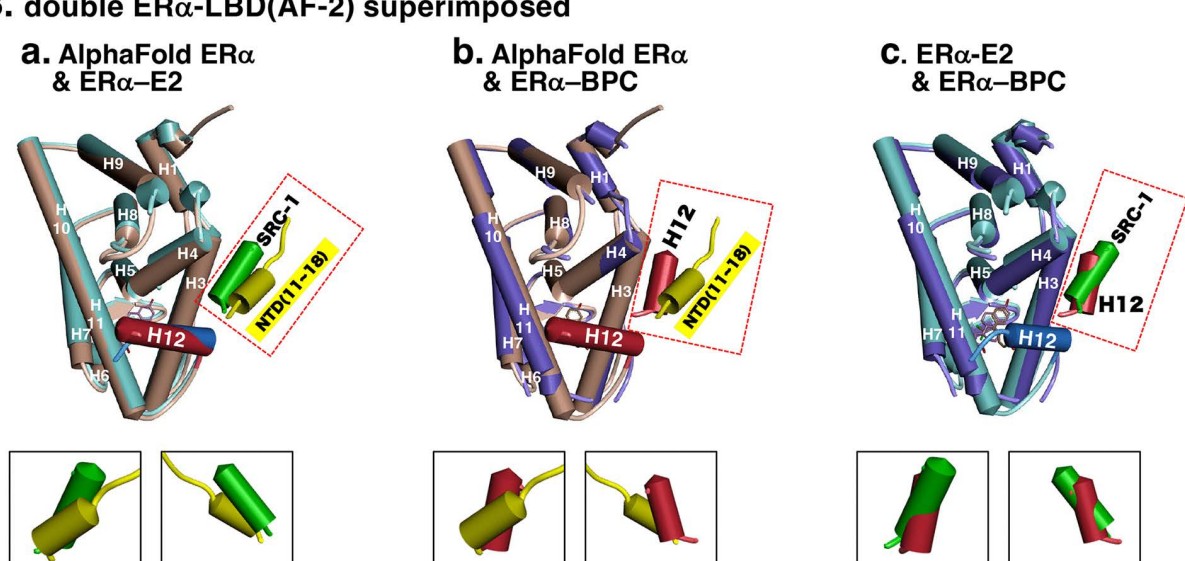

**Fig 7. Orthographic view of superimposed α-helix 12 (H12) in the antagonist positioning on the molecular surface of the ERα-LBD(AF-2) domain.** (**A**) Simplex α-helix structures on the molecular surface of the C-terminal ERα-LBD(AF-2) domain: (**a**) AlphaFold-predicted NTD(11–18)-coupled LBD derived from full-length ERα (P03372), (**b**) SRC-1 NR2 peptide (SRC-1)-coupled LBD in complex with E2 (from PDB 3UUD), and (**c**) the H12 α-helix peptide in the antagonistic position of LBD in complex with BPC (from PDB 3UUC). (**B**) Superimposed structures of the C-terminal ERα-LBD(AF-2) domain: (**a**) AlphaFold-predicted NTD(11–18)-coupled LBD and SRC-1-coupled LBD in complex with E2, (**b**) AlphaFold-predicted NTD(11–18)-coupled LBD and H12 in the antagonistic position of LBD in complex with BPC, and (**c**) SRC-1-coupled LBD in complex with E2 and H12 in the antagonistic position of LBD in complex with BPC. The downward views, from points rotated 120° and 240° apart, of a pair of α-helices are surrounded by a dotted line.

It is now evident that NTD(AF-1) protects ERα from inactivation by BPAF and BPC by interacting with LBD(AF-2). The observations that (i) BPAF and BPC are fully active for full-length ERα and (ii) BPAF and BPC are almost completely inactive for desNTD(AF-1)-ERα underscore the importance of NTD(AF-1) for the wild-type full-length ERα. This importance is further supported by the direct interaction between the NTD(AF-1) and LBD(AF-2) domains. The inactivity of BPAF and BPC for desNTD(AF-1)-ERα raises the question of why and how these bisphenols are rendered inactive.

## Binding characteristics of BPAF and BPC to desNTD(AF-1)-ERα

The 'agonist-to-antagonist functional conversion' induced by BPAF and BPC in ERα occurred due to the removal of NTD(AF-1) from full-length ERα. Therefore, it appears that NTD(AF-1) removal renders the receptor binding of BPAF and BPC ineffective in activating desNTD(AF-1)-ERα. The interaction between NTD(AF-1) and LBD(AF-2) occurs on the molecular surface of ERα-LBD(AF-2), whereas the interaction between the ligand-binding pocket (ERα-LBP) and BPAF/BPC occurs within ERα-LBD(AF-2), near the β-sheet bottom. Therefore, it is essential to identify the structural changes that occur on the ERα-LBD(AF-2) molecular surface and to analyze how these structural changes are induced after BPAF/BPC binding to the ligand-binding site.

BPA was fully active for both full-length ERα and desNTD(AF-1)-ERα. It contains no halogens and can activate desNTD(AF-1)-ERα, albeit considerably weaker than E2 (Fig 2Bb), whereas the halogen-containing BPAF and BPC cannot activate desNTD(AF-1)-ERα. These observations strongly suggest that the inactivity of BPAF and BPC is due to the presence of the halogens fluorine (F) and chlorine (Cl), respectively. BPAF is a structural analog of BPA in which the two methyl ($CH_3$) groups in BPA are replaced with trifluoromethyl ($CF_3$) groups (Fig 1C). BPC contains two Cl atoms in the 2,2-dichloroethylene ($Cl_2C=C<$) group instead of the isopropyl ($(CH_3)_2$-$C<$) group in BPA (Fig 1C). The question is whether these halogen interactions with amino acid residues in the binding pocket affect the interaction between coactivators and LBD(AF-2).

In 2012, Delfosse et al. [41] reported the X-ray crystal structure of ERα-LBD in complex with E2, BPA, BPAF, and BPC. In general, the free LBD(AF-2) domain is used for crystallization of the complex between the NR and the ligand. However, the present study revealed that ERα, especially with BPAF and BPC, changes its mode of receptor activity depending on the presence or absence of the N-terminal NTD(AF-1). This observation suggests that the receptor-binding mode of BPAF and BPC differs between full-length ERα and desNTD(AF-1)-ERα, and also between the agonists (E2 and BPA) and the antagonists (BPAF and BPC). Thus, we decided to carefully review the reported X-ray crystal structures, analyzing them using Discovery Studio software. The structural data were downloaded from the PDB databank.

For crystallization of the protein complex, purified ERα-LBD with a Tyr537Ser point mutation was mixed with $E_2$, BPA or BPAF, and SRC-1 NR2 [41]. SRC-1 NR2, the LXXLL motif-containing NR box2 peptide of SRC-1, was used to stabilize ERα-LBD's H12 in the agonist conformation. By contrast, wild-type ERα-LBD was mixed with BPC, but without SRC-1 NR2 [41]. As shown in S1 Fig, distinct binding modes and H12 conformations were confirmed in their respective crystal complexes with ERα-LBD for BPA, BPAF, and BPC. Two different binding modes of the bisphenol-backbone structure $HO-C_6H_4-C-C_6H_4-OH$ were observed. All of these bisphenols exhibited a characteristic network of hydrogen bonds involving Glu353, $H_2O$, and Arg394 for the hydroxyl group OH in phenol ring A (S1 Fig) (Table 4). A similar hydrogen bond network was observed for the phenolic OH group in the agonist E2 and the antagonist 4-OHT [42] (S1 Fig) (Table 4). 4-OHT was crystallized with Tyr537Ser-ERα-LBD without SRC-1 NR2 [42].

The hydroxyl group of phenol ring B formed a hydrogen bond with His524, incorporating BPA into the BPA-like binding mode (S1C Fig) (Table 4). On the other hand, the hydroxyl group of BPC formed a hydrogen bond with Thr347, creating the BPC-like binding mode (S1F Fig). Notably, BPAF exhibited both binding modes—the BPA-like and BPC-like modes (S1D and S1E Fig)— in each monomer of the ERα-LBD dimer, with a nonrandom distribution (Table 4).

## Structural effects of BPAF and BPC on binding to desNTD(AF-1)-ERα

In the agonist-bound structure, α-helix No. 12 (H12) in LBD(AF-2) occupies the ligand-binding pocket, forming a lid over it. As a result of this binding, H12 provides the hydrophobic coactivator-binding site on the molecular surface. For this reason, H12 is called activation function 2 (AF-2). This H12 conformation is observed for LBD(AF-2) complexed with E2 and BPA (Fig 8), and Delfosse et al. referred to this H12 positioning as "agonist BPA-like positioning," which induces the recruitment of coactivators to the AF-2 surface of ERα-LBD(AF-2) [41]. In contrast, classical antagonists such as 4-OHT

**Table 4. Characteristics of the X-ray crystal structures of the complexes between ERα-LBD(AF-2) and E2, 4-OHT, BPA, BPAF, and BPC.**

| Compound | E2 | 4-OHT | BPA | BPAF | | BPC |
|---|---|---|---|---|---|---|
| PBD ID code | 3UUD[a] | 7UJ8[a] | 3UU7[a] | 3UUA[a] | | 3UUC[a] |
| Amino acid residues binding to the OH group of phenol ring A | Arg394 $H_2O$ Glu353 (17β-OH) | Arg394 $H_2O$ Glu353 (4-OH) | Arg394 $H_2O$ Glu353 | Arg394 $H_2O$ Glu353 | Arg394 $H_2O$ Glu353 | Arg394 $H_2O$ Glu353 |
| Amino acid residues binding to the OH group of phenol ring B | His524 (3-OH) | none | His524 | His524 | Thr347 | Thr347 |
| Conformation of α-helix 12 (H12) | agonist positioning | *antagonist* positioning | agonist positioning | agonist positioning | agonist positioning | *antagonist* positioning |
| Complexing peptide in crystallization | SRC-1 NR2 peptide | none | SRC-1 NR2 peptide | SRC-1 NR2 peptide | SRC-1 NR2 peptide | none |

[a]The analyzed crystal structure contains a Tyr537Ser point mutation in human ERα-LBD. The ERα-LBD structure complexed with BPC is wild type.

The crystal structures of 3UU7, 3UUA, 3UUC, and 3UUD were reported by Delfosse et al. [41], whereas 7UJ8 was reported in [42].

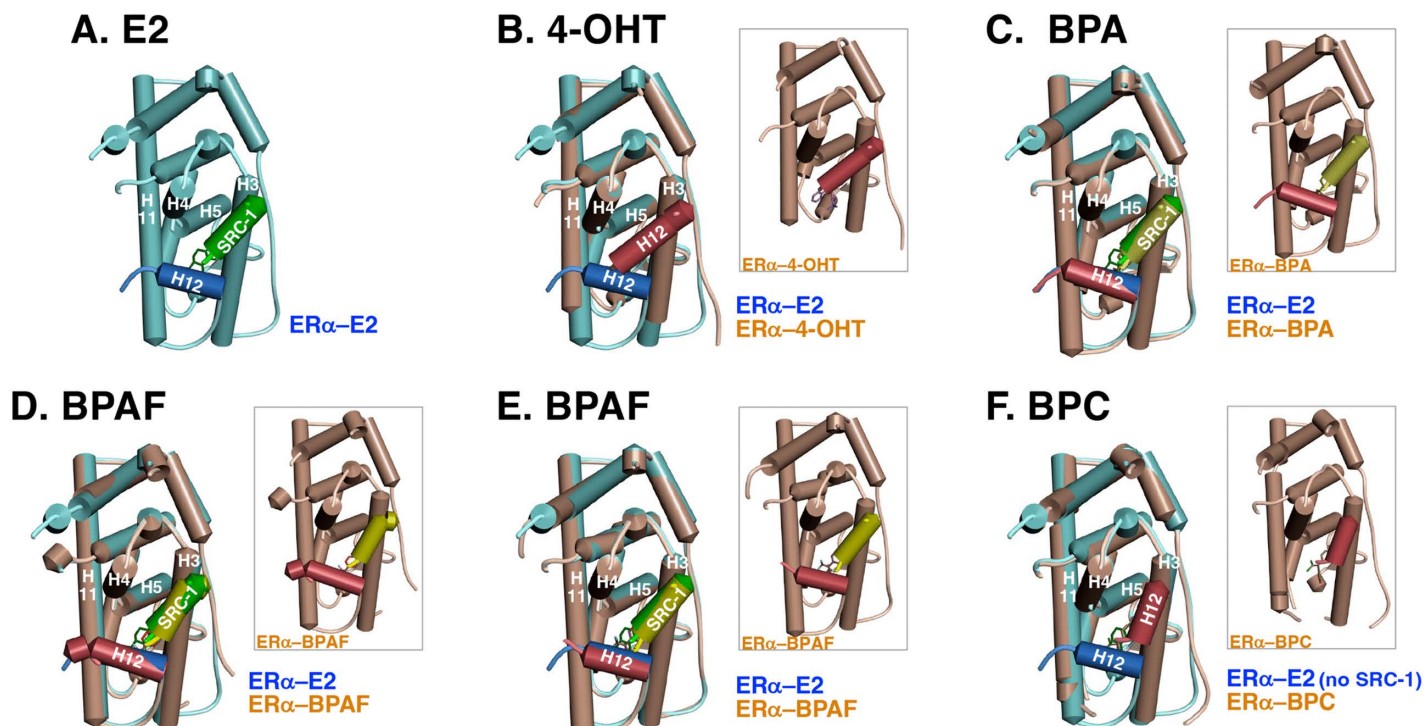

**Fig 8. Schematic of the 3D structure of ERα-LBD(AF-2) in complex with E2 and superimposed structures.** (A) ERα-LBD(AF-2) in complex with E2 and SRC-1 NR2 peptide superimposed with ERα-LBD(AF-2) in complex with 4-OHT (**B**), BPA (**C**), BPAF (**D and E**), and BPC (**F**). SRC-1 NR2 peptide was removed from the structures of E2-complexed ERα-LBD(AF-2) prior to superimposition with ERα-LBD(AF-2) in complex with 4-OHT (**B**) and BPC (**F**). ERα-LBD(AF-2) in complex with E2, 4-OHT, BPA, and BPAF is a Tyr537Ser mutant, whereas that in complex with BPC is wild type. Structures were visualized and analyzed using Discovery Studio software from the PDB (ID codes: (**A**) 3UUD, (**B**) 7UJ8, (**C**) 3UU7, (**D and E**) 3UUA, and (**F**) 3UUC). Structures enclosed in a rectangle are simplex ERα-LBD(AF-2) (brown) in complex with the respective compound.

[42] remodel the ERα-LBD(AF-2) conformation by repositioning H12 to occupy another AF-2 site, thereby blocking coactivator recruitment. This alternative H12 conformation was observed for LBD(AF-2) complexed with BPC and was therefore referred to as "antagonist BPC-like positioning" (Fig 8B and 8F).

As described above, BPAF bound to the ligand-binding site of ERα-LBD(AF-2) exhibited both BPA-like and BPC-like binding modes [41]. When BPAF adopted the BPA-like binding mode, H12 assumed an agonist conformation (Fig 8D) [41]. Notably, even when BPAF adopted the BPC-like binding mode, H12 remained in an agonist conformation (Fig 8E). These observations indicate that, regardless of whether BPAF binds in a BPA-like or BPC-like mode, H12 adopts the same agonist conformation.

If an antagonist conformation of H12 were an absolute requirement for antagonist function, BPAF would not be expected to act as an antagonist. However, BPAF functioned as an antagonist toward desNTD(AF-1)-ERα, suggesting that H12 in the desNTD(AF-1)-ERα–BPAF complex may adopt an antagonist conformation. Consistent with this interpretation, H12 of ERα-LBD(AF-2) complexed with BPC was found to be in an antagonist conformation (Fig 8F), and BPC functioned as an antagonist toward desNTD(AF-1)-ERα. Together, these findings suggest that the conformation of H12 in the BPAF-bound receptor differs between full-length ERα and desNTD(AF-1)-ERα.

## Study on X-ray crystal structures of ERα-LBD complexed with bisphenol

During the analysis of bisphenol binding modes, along with the analysis of agonist/antagonist conformations of the H12 α-helix, we observed that the AlphaFold-predicted NTD(11–18) α-helix peptide (Fig 7Aa), the X-ray crystal-complexed SRC-1 NR2 peptide (Fig 7Ab), and the H12 α-helix peptide in the antagonist conformation (Fig 7Ac) are situated at very similar positions on the molecular surface of ERα-LBD(AF-2). Superimposition of two of these structures, as shown in Fig 7B, revealed some combinations standing side by side, with slight twisting and contact. These include the combinations between the AlphaFold-predicted NTD(11–18) α-helix peptide and the complexed SRC-1 NR2 peptide (Fig 7Ba), and between the AlphaFold-predicted NTD(11–18) α-helix peptide and H12 in the antagonist conformation (Fig 7Bb).

Fig 7Ba shows that the combination between the AlphaFold-predicted NTD(11–18) α-helix peptide and the X-ray cindrystal-complexed SRC-1 NR2 peptide is feasible. This would ensure that ERα-NTD(AF-1) recruits the SRC coactivator protein(s) by being adjacent to ERα-LBD(AF-2) [1,12]. The close proximity of this α-helix peptide and the SRC-1 NR2 peptide appears to reflect a true function of the NTD(11–18) α-helix peptide. Each peptide interacts with amino acid residues on the surface of LBD(AF-2) (S2 Fig). In particular, the SRC-1 NR2 peptide is involved in a complex network of interactions, including nine hydrogen bonds and seven hydrophobic bonds (S2B Fig), indicating that this SRC-1 NR2 peptide binds very tightly to H3, H4, and H12 of LBD(AF-2). The AlphaFold-predicted NTD(11–18) α-helix peptide is also involved in a network of interactions (hydrogen bonds and hydrophobic bonds) (S2A Fig).

The formation of the latter combination, namely, the superimposition between the NTD(11–18) α-helix peptide and the H12 peptide in the antagonist conformation (Fig 7Bb), is another important structural characteristic. The compatibility of H12 in the antagonist confirmation with the NTD(11–18) α-helix indicates that H12 can adopt both the agonist and antagonist conformations on the ERα-LBD(AF-2) receptor surface. In other words, the interaction or binding of ERα-NTD(AF-1) to ERα-LBD(AF-2) does not hinder either of H12's agonist/antagonist conformations and may even stabilize the H12 antagonist conformation.

It is thus likely that the NTD(11–18) α-helix peptide of the ERα-NTD(AF-1) domain interacts functionally with either the SRC-1 NR2 peptide or H12 in the antagonist positioning. These structural characteristics of the NTD(11–18) α-helix peptide likely constitute the molecular basis for ERα function. Moreover, the H12 α-helix peptide in the antagonist positioning is also involved in a network of interactions (hydrogen bonds and hydrophobic bonds) (S2C Fig).

## SRC-1 and H12 peptides are incompatible in the antagonist conformation

The remaining combination is the superimposition of the crystallographically complexed SRC-1 NR2 peptide and H12 in the antagonist positioning. However, these two α-helix peptides overlap considerably, such that they sterically encroach on

one another (Fig 7Bc). This suggests that coactivator recruitment and the antagonist positioning of H12 never occur simultaneously, indicating that transcriptional activation and inhibition cannot coincide. We noted that H12 of the ERα-LBD in complex with BPAF adopts the agonist positioning (Fig 8E), even when BPAF binds in a BPC-like binding mode within the ligand-binding pocket (S1E and S1F Fig). We suspect that H12 remained in the agonist positioning because the crystallographically bound SRC-1 NR2 peptide occupies the position that H12 would otherwise adopt (Fig 8E).

In contrast, H12 of the ERα-LBD in complex with BPC adopts the antagonist positioning (Fig 8F). Notably, in this X-ray crystal structure, crystallization was performed without SRC-1 NR2 peptide. In that case, H12 peptide can readily occupy the antagonist positioning. The H12 of ERα-LBD in complex with 4-OHT was also in an antagonist positioning in the absence of SRC-1 NR2 peptide (Fig 8B).

We again question what would happen if ERα-LBD in complex with BPAF were crystallized without SRC-1 NR2 peptide. It is possible that H12 would adopt the antagonist positioning. If ERα-LBD in complex with BPAF were crystallized without SRC-1 NR2 peptide, H12 might adopt the antagonist positioning. It is also possible that, if ERα-LBD in complex with BPC were crystallized with SRC-1 NR2 peptide, H12 would remain in the agonist positioning.

## Perspectives on the inactivation of desNTD(AF-1)-ERα by BPAF and BPC

We confirmed that BPAF and BPC are fully active agonists for the wild-type full-length ERα but inactive antagonists for desNTD(AF-1)-ERα. E2 was active and an agonist for both full-length ERα and desNTD(AF-1)-ERα, whereas 4-OHT and ICI were inactive antagonists for both full-length ERα and desNTD(AF-1)-ERα. The structural difference between full-length ERα and desNTD(AF-1)-ERα is the presence or absence of the N-terminal NTD(AF-1) domain. This difference results in the presence or absence of the interaction between the N-terminal NTD(AF-1) domain and the C-terminal LBD(AF-2) domain.

H12 of full-length ERα is typically located at the top of the ligand-binding pocket, where it contributes to the hydrophobic core through packing with neighboring H11. ERα antagonists 4-OHT and ICI displace H12 from this agonist positioning to the antagonist positioning by disrupting the hydrophobic core or via bulky hydrophobic groups present in their molecules (Fig 1Cb and 1Cc) [43–45]. It remains unclear whether the interaction between NTD(AF-1) and LBD(AF-2) breaks down concurrently.

Since BPAF and BPC do not possess such bulky hydrophobic groups (Fig 1Ce and 1Cf), it is unlikely that these bisphenols can disrupt the hydrophobic core in full-length ERα and shift H12 to the antagonist positioning. The interaction between the N-terminal NTD(AF-1) domain and the C-terminal LBD(AF-2) domain may contribute to maintaining the hydrophobic core. This interaction likely explains why BPAF and BPC function as agonists for the full-length ERα.

Our recent investigation to identify similar interactions between NTD(AF-1) and LBD(AF-2) has revealed at least three interactions, including one involving the NTD(11–18) α-helix peptide. These investigations are under way and will be reported in due course. It is likely that these interactions stabilize the transcription activation function of estrogens, collaborating with the C-terminal LBD(AF-2) domain. In addition, these interactions may prevent H12 from adopting the antagonist conformation even after BPAF or BPC binding to the ligand-binding pocket, thereby contributing to the agonist activity of BPAF and BPC.

In the shortened ERα derivative desNTD(AF-1)-ERα in which the N-terminal NTD(AF-1) domain was completely removed, interactions between NTD(AF-1) and LBD(AF-2) are absent. Thus, without additional factors hindering the placement of H12 in the antagonist positioning, BPAF and BPC can eventually disrupt the hydrophobic core between H11 and H12, and then shift H12 to the antagonist positioning (Fig 8F). The driving force behind this translocation of H12 is likely the halogen bonding by $CF_3$ in BPAF and the Cl atom in BPC molecules, although the specific halogen bonding interactions responsible for the destruction of the hydrophobic core remain unclear. This motive power of halogen bonding is fully displayed for ERα only when NTD(AF-1) is removed. Thus, further investigation is necessary to elucidate the complete picture of the interaction(s) between NTD(AF-1) and LBD(AF-2) in ERα.

## Conclusions

Removal of the N-terminal domain NTD(AF-1) from wild-type ERα retains approximately 65% of the transcriptional activation activity of full-length ERα. However, this removal converts ERα-active BPAF and BPC into inactive, antagonistic compounds. These results strongly suggest that NTD(AF-1) plays a function-determining role, specifically by influencing the conformation of the H12 α-helix peptide of the C-terminal LBD(AF-2), shifting it from an agonist to an antagonist positioning. The driving force for this H12 translocation from the agonist positioning to the antagonist positioning is likely due to the specific halogen bonding by the $CF_3$ groups in BPAF and the Cl atoms in BPC. These halogen bondings eventually appear to destruct the hydrophobic bondings between the H11 and H12. However, the effect of halogen bondings to release H12 to the antagonist positioning was shown only when NTD(AF-1) was removed, and therefore it is strongly suspected that the interaction between N-terminal NTD(AF-1) and C-terminal LBD(AF-2) works to block the effect of halogen bondings.

In conclusion, further investigation is necessary to elucidate the complete picture of the actual formations of halogen bondings, and also of the actual molecular mechanism of the H12-translocation. We will expose the structural characteristics of BPAF and BPC in their halogen bondings with the ligand-binding site of ERα. In particular, elucidating the role of NTD(AF-1) in regulating H12 translocation is of significant interest. The present results may provide insight into the NTD(AF-1)-specific molecular mechanism underlying the antagonistic effects of BPAF and BPC on desNTD(AF-1)-ERα, and conversely, the agonistic effects of BPAF and BPC on full-length ERα.

## Supporting information

**S1 Fig. ERα-LBD receptor-binding mode of a series of ligands.**
(PDF)

**S2 Fig. Interaction networks of α-helix structures on the molecular surface of the C-terminal ERα-LBD(AF-2) domain.**
(PDF)

**S1 Table. Data for Schild plot analysis of the antagonist BPAF and BPC with desNTD(AF-1)-ERα.**
(PDF)

**S2 Table. Data for Schild plot analysis of 4-hydroxytamoxifen (4-OHT) with full-length ERα and desNTD(AF-1)-ERα.**
(PDF)

**S3 Table. Data for Schild plot analysis of the antagonist ICI 182,780 (ICI) with full-length ERα and desNTD(AF-1)-ERα.**
(PDF)

## Author contributions

**Conceptualization:** Yasuyuki Shimohigashi.

**Data curation:** Xiaohui Liu, Yasuyuki Shimohigashi.

**Formal analysis:** Xiaohui Liu, Yasuyuki Shimohigashi.

**Funding acquisition:** Xiaohui Liu, Yasuyuki Shimohigashi.

**Methodology:** Xiaohui Liu, Miki Shimohigashi, Yasuyuki Shimohigashi.

**Project administration:** Xiaohui Liu, Miki Shimohigashi, Yasuyuki Shimohigashi.

 

**Resources:** Xiaohui Liu, Yasuyuki Shimohigashi.

**Software:** Xiaohui Liu.

**Supervision:** Xiaohui Liu, Yasuyuki Shimohigashi.

**Validation:** Xiaohui Liu, Miki Shimohigashi, Yasuyuki Shimohigashi.

**Writing – original draft:** Xiaohui Liu, Yasuyuki Shimohigashi.

**Writing – review & editing:** Xiaohui Liu, Miki Shimohigashi, Yasuyuki Shimohigashi.

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
