## [Decision Letter · Decision Letter 0]

9 Apr 2026

PONE-D-26-08447Bisphenol BPAF and BPC are agonists for estrogen receptor ERα but antagonists for N-terminal domain-lacking ERαPLOS One

Dear Dr. Liu,

Thank you for submitting your manuscript to PLOS ONE. After careful consideration, we feel that it has merit but does not fully meet PLOS ONE’s publication criteria as it currently stands. Therefore, we invite you to submit a revised version of the manuscript that addresses the points raised during the review process.

We look forward to receiving your revised manuscript.

Kind regards,

Sayyed Mohammad Hadi Alavi

Academic Editor

PLOS One

Journal Requirements:

“This work was supported by the Japan Society for the Promotion of Science (JSPS) KAKENHI grant nos. 25K15462, 22K12395, 19K12340, 15K00557, and 25740024 (to X.L.) and 22221005 and 15H01741 (to Y.S.). This work was also supported in part by a Health and Labour Sciences Research Grant for Research on the Risk of Chemical Substances from the Ministry of Health, Labor, and Welfare of Japan, under contract/grant no. H20-Chemistry-General-003 (to Y.S.).”

“This work was supported by the Japan Society for the Promotion of Science (JSPS) KAKENHI grant nos. 25K15462, 22K12395, 19K12340, 15K00557, and 25740024 (to X.L.) and 22221005 and 15H01741 (to Y.S.). This work was also supported in part by a Health and Labour Sciences Research Grant for Research on the Risk of Chemical Substances from the Ministry of Health, Labor, and Welfare of Japan, under contract/grant no. H20-Chemistry-General-003 (to Y.S.).”

“This work was supported by the Japan Society for the Promotion of Science (JSPS) KAKENHI grant nos. 25K15462, 22K12395, 19K12340, 15K00557, and 25740024 (to X.L.) and 22221005 and 15H01741 (to Y.S.). This work was also supported in part by a Health and Labour Sciences Research Grant for Research on the Risk of Chemical Substances from the Ministry of Health, Labor, and Welfare of Japan, under contract/grant no. H20-Chemistry-General-003 (to Y.S.).”

5. We note that your Data Availability Statement is currently as follows: “All relevant data are within the manuscript and its Supporting Information files”

6.  We notice that your supplementary figures are uploaded with the file type 'Figure'. Please amend the file type to 'Supporting Information'. Please ensure that each Supporting Information file has a legend listed in the manuscript after the references list.

Additional Editor Comments:

Dear Dr. Xiaohui Liu

Thank you very much for your submission to PLOS ONE. Your MS was reviewed by two reviewers with expertise in Chemical Biology/Receptor Biology. Both provided very valuable comments that need your careful consideration upon revision. In addition to their comments, please find my comments below to include into revision. I am looking forward to receiving your revision.

Very best regards

AE COMMENTS:

- Please follow the PLOS ONE guidelines to organize the MS.

- Please clarify why P < 0.001 was considered for significant difference; usually P < 0.05 is taken into account.

- Did you check the normal distribution and homogeneity of variance for t-Test?

- Was not it possible to show the significant differences on Figures or in Tables?

- Please add number of experiments performed for each assay, both in the method, in the legend of figures, and in the title of Tables.

- For funding: Please state what role the funders took in the study. If the funders had no role, please state: ""The funders had no role in study design, data collection and analysis, decision to publish, or preparation of the manuscript.""

- Please clarify "Competing Interests Statement"

- Please provide Ethical statement.

Reviewer's Responses to Questions

**Comments to the Author**

1. Is the manuscript technically sound, and do the data support the conclusions?

Reviewer #1: Yes

Reviewer #2: Yes

2. Has the statistical analysis been performed appropriately and rigorously? 

Reviewer #1: Yes

Reviewer #2: Yes

3. Have the authors made all data underlying the findings in their manuscript fully available?

Reviewer #1: No

Reviewer #2: Yes

4. Is the manuscript presented in an intelligible fashion and written in standard English?

Reviewer #1: No

Reviewer #2: Yes

5. Review Comments to the Author

Reviewer #1: General comment :

1. All the figures, including those as supplementary ones, have to be presented with a better resolution.

Minor comment :

1. Line 110, Begin the sentence with: Indeed,

2. Line 137, Remove the last dot.

3. For clarity, add for Figure 1C, add below the structures the corresponding E2, 4-OH, ICI, BPA, BPAF, BPC

4. Line 250 to 252, simplify the sentence. Remove the semicolon in line 251 and write again the sentence

5. Line 331, “eventually reaching baseline”. Remove this part and be more precise.

6. Remove sentence from line 445 to 446.

7. For Figure 7, indicate the helices number from 1 to 12

Reviewer #2: This manuscript describes and investigation of binding and functional properties of bis-phenol A (BPA), the bis-trifluoromethylated analog (BPAF) and a gem-dichloroethylene analog (BPC) with wild-type Estrogen Receptor alpha and the modified protein that lacks the N-terminal domain [NTD(AF-1)]. The major findings are that the modified protein retains a significant amount of transcriptional activity relative to the full-length protein, however, while the simple parent compound BPA is an agonist for both proteins, the halogenated small molecules behave as antagonists in the absence of the N-terminal domain. The paper is written clearly, the experiments are described in sufficient detail, and the references are appropriate. This concluding statement is unsatisfactory and needs further elaboration: “The driving force for this H12 translocation is likely due to the halogen atoms present in BPAF and BPC, and its effect is only fully observed upon NTD(AF-1) removal.” The significance of these findings could be clarifies and strengthened for the reader by addressing several related questions. What is the broader context of these specific results of bisphenol-related environmental estrogens within the classical example of estrogen receptor alpha (ERα) as a ligand-dependent transcription factor and why should we care? What are the major structural/steric differences between these compounds evaluated and how are the halogens involved, given their ability to modify properties such as acidity through induction, and the capacity of fluorinated alkanes to act as weak hydrogen bond acceptors? Are there known differences in the biological or toxicological activity profiles of the compounds? The fact that there are numerous clinically significant mutations in the ER’s ligand-binding domain that are relevant for cancer raises the potential here to include discussion of known mutations and structural variants in or involving the N-terminal domain (NTD) that contribute to pathological conditions. One other minor comment is that in Table 1 the column titled relative “potency” should be changed to “affinity”.

6. PLOS authors have the option to publish the peer review history of their article (what does this mean?). If published, this will include your full peer review and any attached files.

Reviewer #1: No

Reviewer #2: No

---

## [Author Response · Author response to Decision Letter 1]

28 Apr 2026

Dear Dr. Alavi and Reviewers,

Thank you very much for giving us the opportunity to revise our manuscript submitted to PLOS ONE. We appreciate the constructive feedback provided by the editors and reviewers.

We have addressed all comments and suggestions provided by the Academic Editor and the two Reviewers. Our detailed point-by-point responses are provided in the uploaded file entitled "Response to Reviewers."

Key revisions in this resubmission include clarification of significance, in-depth discussion, statistical and visual updates, and improved data availability.

We believe that these revisions have significantly strengthened the manuscript and hope that it is now acceptable for publication in PLOS ONE. Thank you for your continued consideration.

Sincerely yours,

Corresponding Authors:

Dr. Xiaohui LIU

Prof. Yasuyuki SHIMOHIGASHI

---

## [Decision Letter · Decision Letter 1]

14 May 2026

Bisphenol BPAF and BPC are agonists for estrogen receptor ERα but antagonists for N-terminal domain-lacking ERα

PONE-D-26-08447R1

Dear Dr. Liu,

We’re pleased to inform you that your manuscript has been judged scientifically suitable for publication and will be formally accepted for publication once it meets all outstanding technical requirements.

Kind regards,

Sayyed Mohammad Hadi Alavi, PhD

Academic Editor

PLOS One

Additional Editor Comments (optional):

Dear Dr Liu

Thank you for your submission to PLOS ONE. You revision has been accepted for publication.

Very best regards

Hadi Alavi

Reviewers' comments:

Reviewer's Responses to Questions

**Comments to the Author**

1. If the authors have adequately addressed your comments raised in a previous round of review and you feel that this manuscript is now acceptable for publication, you may indicate that here to bypass the “Comments to the Author” section, enter your conflict of interest statement in the “Confidential to Editor” section, and submit your "Accept" recommendation.

Reviewer #1: All comments have been addressed

Reviewer #2: All comments have been addressed

2. Is the manuscript technically sound, and do the data support the conclusions?

Reviewer #1: Yes

Reviewer #2: Yes

3. Has the statistical analysis been performed appropriately and rigorously? 

Reviewer #1: Yes

Reviewer #2: Yes

4. Have the authors made all data underlying the findings in their manuscript fully available?

Reviewer #1: Yes

Reviewer #2: Yes

5. Is the manuscript presented in an intelligible fashion and written in standard English?

Reviewer #1: Yes

Reviewer #2: Yes

6. Review Comments to the Author

Reviewer #1: General comment :

1. The authors are carefully addressed my comments and improved significantly their revised manuscript.

Especially :

a) The labelling of compound name for the structures is now inserted in Fig 1C.

b) The resolution of figures is clearly better.

The authors have further stated their major findings and indicate the limit of their study, which was required.

Minor Comment:

- Page 3, line 80, Remove Cryo-EM in parenthesis as it appears one time in the manuscript

Reviewer #2: As stated in part 1. All comments have been addressed. "If the authors have adequately addressed your comments raised in a previous round of review and you feel that this manuscript is now acceptable for publication, you may indicate that here to bypass the “Comments to the Author” section, enter your conflict of interest statement in the “Confidential to Editor” section, and submit your "Accept" recommendation." The electronic review form is indicating the the minimum character count is not met, and the form will not proceed to submission, so I am adding this additional text to meet the minimum character count.

7. PLOS authors have the option to publish the peer review history of their article (what does this mean?). If published, this will include your full peer review and any attached files.

Reviewer #1: No

Reviewer #2: No

---

## [Editor Report · Acceptance letter]

PONE-D-26-08447R1

PLOS One

Dear Dr. Liu,

I'm pleased to inform you that your manuscript has been deemed suitable for publication in PLOS One. Congratulations! Your manuscript is now being handed over to our production team.

Kind regards,

on behalf of

Dr. Sayyed Mohammad Hadi Alavi

Academic Editor

PLOS One